# Optimal Fleet Transition Modeling for Sustainable Inland Waterways Transport

Matteo Giacomo Prina *, Alyona Zubaryeva, Giuseppe Rotondo, Andrea Grotto and Wolfram Sparber

Institute for Renewable Energy, EURAC Research, Viale Druso 1, I-39100 Bolzano, Italy; alyona.zubaryeva@eurac.edu (A.Z.); giuseppe.rotondo@eurac.edu (G.R.); andrea.grotto@eurac.edu (A.G.); wolfram.sparber@eurac.edu (W.S.)
* Correspondence: matteogiacomo.prina@eurac.edu; Tel.: +39-0471055587

**Abstract:** The transition to sustainable waterways transport is imperative in the face of environmental and climate challenges. Local lakes, often overlooked, play a significant role in regional transportation networks and ecosystems. This study focuses on Orta lake, Italy, and aims to facilitate its transition to sustainable inland waterways transport by substituting its diesel-based fleet with electric vessels. Firstly, a comprehensive market analysis was conducted to understand the available electric vessel models and their technical characteristics. This included parameters such as capacity, range, and charging time. Based on the market analysis, an optimization model was developed to determine the minimum number of electric vessels required to completely replace the existing diesel-based fleet. This model considers various constraints and objectives, such as meeting transport demand, minimizing the number of vessels, and reducing environmental impact. The developed model was then applied to the case study of Orta lake using the collected market data. The results indicate an optimal fleet configuration and provide insights into the feasibility and implications of the transition. This study contributes to the growing body of knowledge on sustainable inland waterways transport and offers a methodology that can be replicated and adapted for other local lakes or maritime settings.

**Keywords:** sustainable inland waterways transport; electric ships; fleet transition; optimization model

## 1. Introduction

The global transportation sector is a significant contributor to greenhouse gas emissions, accounting for approximately 23% of global energy-related $CO_2$ emissions in 2019 [1]. Among the various modes of transport, maritime transport was responsible for about 2.89% of global greenhouse gas emissions in 2018 [2]. In EU, maritime transport accounted for 13.5% of EU transport emissions in 2019 [3]. While these figures primarily represent oceanic shipping, inland water transport, including local lakes, also contributes to these emissions and thus cannot be overlooked in the transition to sustainable transport systems.

In line with the global commitment to mitigate the effects of climate change, the European Union has embarked on an ambitious path towards sustainable development through its European Green Deal and the 2030 Climate Target Plan [4]. These strategic initiatives mandate an aggressive reduction of greenhouse gas (GHG) emissions, targeting at least a 55% reduction by 2030 compared to the 1990 levels, and strive for climate neutrality by 2050. All sectors of the economy, including maritime transport, are expected to contribute towards this ambitious environmental target [5].

To facilitate this objective within the maritime sector, the FuelEU Maritime initiative has been implemented [6]. As outlined in its provisions, this initiative aims to gradually decarbonize maritime transport by reducing the GHG intensity of the energy used onboard vessels. The transition towards greener and more sustainable energy sources for vessels is a critical step in this initiative, with the ultimate aim of completely decarbonizing the sector. By driving regulatory changes and promoting innovations in sustainable vessels, the

FuelEU Maritime initiative forms a cornerstone of the European Union's approach towards achieving its climate goals.

The decarbonization of waterborne transport presents numerous challenges, but also opens up a range of opportunities, especially with respect to technological advancements [5]. One such approach to reducing GHG emissions from maritime transport is the incorporation of zero-emission-fueled vessels into existing fleets. This transition to more sustainable energy sources is gaining increasing attention in scientific research and policy-making. The decarbonization of the transportation sector has been widely studied using energy system models. Several articles have examined the impacts of increased electrification on emissions and costs. For example, Bellocchi et al. [7] used EnergyPLAN [8] to analyze different levels of private vehicle electrification. They found electrification reduces costs and emissions, but grids may require capacity expansion. Similarly, Tamba et al. [9] showed that electrifying India's roads lowered fuel costs and supported climate change mitigation. However, Tamor et al. [10] demonstrated replacing all US personal vehicles with battery electric ones would only cut transportation emissions by about half due to electricity generation impacts. Sector-coupling between transport and electricity is very relevant and is a challenge for the energy system modeling research field [11]. These approaches have also been applied to vessels. Groppi et al. [12] developed marginal abatement cost curves for a small island where maritime transport dominated emissions. Electrification reduced emissions at reasonable costs. While prior works focus on roads and ocean-going vessels, local inland water transport has received less attention.

Inland vessels, in particular, are being encouraged to consider a shift towards low-carbon energy sources. Such sources include, but are not limited to, lithium batteries [13], fuel cells [14], ammonia and methanol [15], and wind-assist and sail propulsion [16]. Several studies [13–16] have confirmed that the use of these clean energy sources to progressively replace traditional diesel engines is a significant trend in the ongoing decarbonization process of vessel power technologies. Each of these technologies brings its own unique set of characteristics and potential benefits, but all align with the broader objective of decarbonization. Perčić et al. [17] assessed diesel-versus electric-battery-powered inland vessels with and without photovoltaics for three vessel types. While electric vessels with photovoltaics had the lowest environmental impact, their life cycle costs were prohibitive for cargo and dredger vessels. However, for passenger vessels, battery electric power with photovoltaics was found to be a cost-effective, low-emission option. This suggests that electrification may be the most feasible strategy for sustainable transition of inland passenger fleets.

This study focuses on the potential for electrifying the fleet at Orta lake, aiming to offer insights and methodologies that could be replicated in other maritime contexts across Europe. Orta lake, located in Italy, is one such local lake where inland waterways transport plays a relevant role for tourists' transportation. The lake's fleet is currently diesel-based, contributing to both local and global environmental impacts. The transition to a more sustainable fleet, specifically electric vessels, is therefore a pressing need. However, such a transition requires careful planning and optimization to ensure feasibility and effectiveness. In case of vessels in canals or lakes (~1000 t), scholars recommend the usage of lithium battery vessel technology with fast charging for short trips [17].

This paper aims to address this problem by developing an optimization model to determine the minimum number of electric vessels required to replace the existing diesel-based fleet entirely. This model is based on a comprehensive market analysis of available electric vessel models and their technical characteristics, such as capacity and range. The model is then applied to the case study of Orta lake, providing insights into the feasibility and implications of the transition. The optimization problem applied to the transition from diesel to electric vessels is done with a primary constraint, namely that the time schedule of vessels arrivals and departures remains unchanged.

The shift towards sustainable alternatives in maritime transport fleets has garnered increasing attention in recent years. This transition is a critical aspect of broader efforts to

reduce greenhouse gas emissions and mitigate climate change. While there is a substantial body of literature on optimizing sustainable transitions for public transport (e.g., buses) routes, similar research focusing specifically on maritime transport is not as extensive and has only begun to emerge more prominently in recent years.

In the topic of sustainable transitions for bus routes, several studies have utilized mixed-integer linear models for optimization. For instance, a study [18] conducted an analysis of multiple bus routes at an urban level in Berlin, Germany, focusing on battery electric buses. The study considered parameters such as individual routes, bus types, and traffic, but did not delve into the impact of high slope variation or longer distances due to the specific topology of the case. Similarly, another study [19] employed a mixed-integer linear cost optimization model to explore the potential for substituting traditional public transportation modes with electric buses. This study focused on identifying the least energy-consuming routes and potential locations for electric bus infrastructure, assuming only constant slope values in a simplified road network. In a different approach, the authors of [20] developed a bi-objective model that minimizes both total carbon emission and operational cost for a mixed fleet of electric and diesel buses in the city of Liuzhou. While this study considered travel distance in their energy consumption model, it did not include slope considerations. A study [21] applied a mixed-integer nonlinear programming model to the transit network of Belleville city, Canada, with a detailed calculation of energy consumption for electric buses. This study primarily considered the impact of traffic intensity, as the slope variation in the case study was low. Another study [22] proposed an optimization approach using a mixed-integer linear model for the electrification of a bus route in Singapore, considering factors such as partial charging and battery degradation.

Rinaldi et al. (2018) [23], Sassi and Oulamara (2017) [24], and Zhang et al. (2021) [25], have considered aspects such as vehicle scheduling for electric and hybrid buses, optimal charging, and the assignment of electric buses to predefined routes.

Lastly, Sparber et al. [26] explored the feasibility of transitioning an entire regional bus fleet in an Alpine region of Italy to zero-emission buses, specifically battery electric buses (BEBs) and fuel cell electric buses (FCEBs). They developed a model based on the vehicle's energy balance, taking into account factors such as distance, altitude, and climate conditions.

In the topic of sustainable transitions for maritime transport routes, while different studies have addressed the network design and vehicle scheduling problem [27–29], few studies have specifically addressed the optimization of passenger vessel services with a focus on zero-emission vessels.

Pratt and Klebanoff (2018) [30] examined the design and feasibility of zero-emission hydrogen-fuel-cell-powered ferries through case studies of five different passenger vessel types. A key finding is that larger, lower speed ferry designs are the most cost-effective starting point for deploying currently available hydrogen fuel cell technology, resulting in lower emissions, capital costs, and operating costs per passenger mile compared to higher speed designs.

Sundvor et al. (2021) [31] focused on the potential for replacing high-speed passenger vessels in Norway with zero-emission solutions such as compressed hydrogen or battery electric zero-emission vessels. They highlight the necessity of route optimization studies to properly support decision makers in choosing the best sustainable maritime solutions.

Havre et al. (2022) [32] focused on the Zero-Emission Vessel Route Planning Problem for planning routes serviced by battery electric vessels. For this scope, they present a mixed-integer programming optimization model which jointly minimizes operator and passenger costs. This model is used to evaluate technology replacement potential and associated costs for two routes in Norway. The results show that transitioning to zero-emission technology increases operational costs due to battery range and charging time constraints, which may require more vessels and result in reduced frequency and increased passenger discomfort.

Nikolopoulos and Boulougouris (2023) [33] presented a simulation-driven robust optimization approach for the design of zero-emission vessels powered by ammonia. Their

methodology uses dynamic voyage simulation integrated with a holistic vessel design model to optimize the design for economic and environmental performance.

The approach presented in this study differs from the ones mentioned above in that it specifically addresses the transition of a diesel-based fleet to an electric one in a local lake setting. The model takes into account various characteristics of the lake, the boats, and the routes, and it updates the status and state of charge of each vessel for each timestep and each route. This allows the model to evaluate the minimum number of electric vessels necessary to completely cover the routes and their timetables that are currently covered by diesel boats. In order to achieve this, a greedy heuristic algorithm is adopted to solve the optimization problem.

The literature applying optimization models specifically to study the transition from diesel-based passenger vessel fleets to electric fleets is very recent and limited. This work provides an original contribution by implementing a greedy heuristic algorithm to determine the minimum number of vessels required and analyzes charging infrastructure needs for a full diesel to electric fleet transition. Given the nascency of research on optimizing electric passenger vessel fleet transitions, this work presents a novel methodology. The simplicity and yet high practical utility of the greedy heuristic algorithm are tailored to this problem. It balances solution quality and speed to provide essential insights into community-level electric fleet transition feasibility and implementation requirements.

The Materials and Methods section will detail the optimization model developed for this study. This model, designed to determine the minimum number of electric vessels required to replace the existing diesel-based fleet, takes into account various parameters such as capacity, range, charging time, and costs of the available electric vessel models. Following this, the Case Study section will introduce Orta lake, the local lake in Italy that serves as the focus of this study. This section will provide an overview of the lake, its current diesel-based fleet, and the specific challenges and considerations related to its transition to sustainable inland waterways transport. The Results section will present the findings obtained from applying the optimization model to the case study of Orta lake. This will include the optimal fleet configuration, the feasibility of the transition, and the implications of the transition. Finally, the Conclusions section will provide conclusive remarks on the study. It will summarize the key findings, discuss their implications for sustainable inland waterways transport, and suggest potential directions for future research in this area.

## 2. Materials and Methods

The optimization model developed in this study aims to determine the minimum number of electric vessels required to replace the existing diesel-based fleet in Orta lake. The model is based on a comprehensive market analysis of available electric vessel models and their technical characteristics, including capacity, range, charging time, and costs.

The model operates in discrete timesteps, with each timestep representing a specific duration (e.g., one hour). For each timestep and each route, the model checks if there is any available vessel with enough battery charge to make the entire trip of the considered route. If this is not the case, the model adds a new vessel. Figure 1 shows the basic steps that compose the greedy heuristic algorithm. The algorithm follows a specific procedure to determine the minimum electric fleet requirements. First, the necessary inputs are gathered, including technical details on the lake, routes, timetables, and vessel specifications. With this critical information in hand, the model can then begin its analysis. The evaluation occurs on a timestep basis, incrementing through the entire planning horizon. At each timestep, the model checks the schedule to see if any vessels are due to depart on their routes from the ports. If a departure is scheduled, the model then looks to see if there is an electric vessel available at that port with enough charge to make the full trip. Availability is determined based on the current status of each vessel already in the system and its own state of charge. The state of charge should be enough to cover the scheduled travel. If no satisfactory electric vessel can be found, this signals the need to add another electric boat to the fleet. Once any necessary vessels are added, the model updates the status of

all vessels to reflect their new positions and charges. This sequential process is repeated for every timestep as the model iterates through the schedule. By methodically stepping through each timestep, checking vessel availability against scheduled departures, and adding new vessels only when required, the model can determine the minimum fleet size to fully replace the diesel boats.

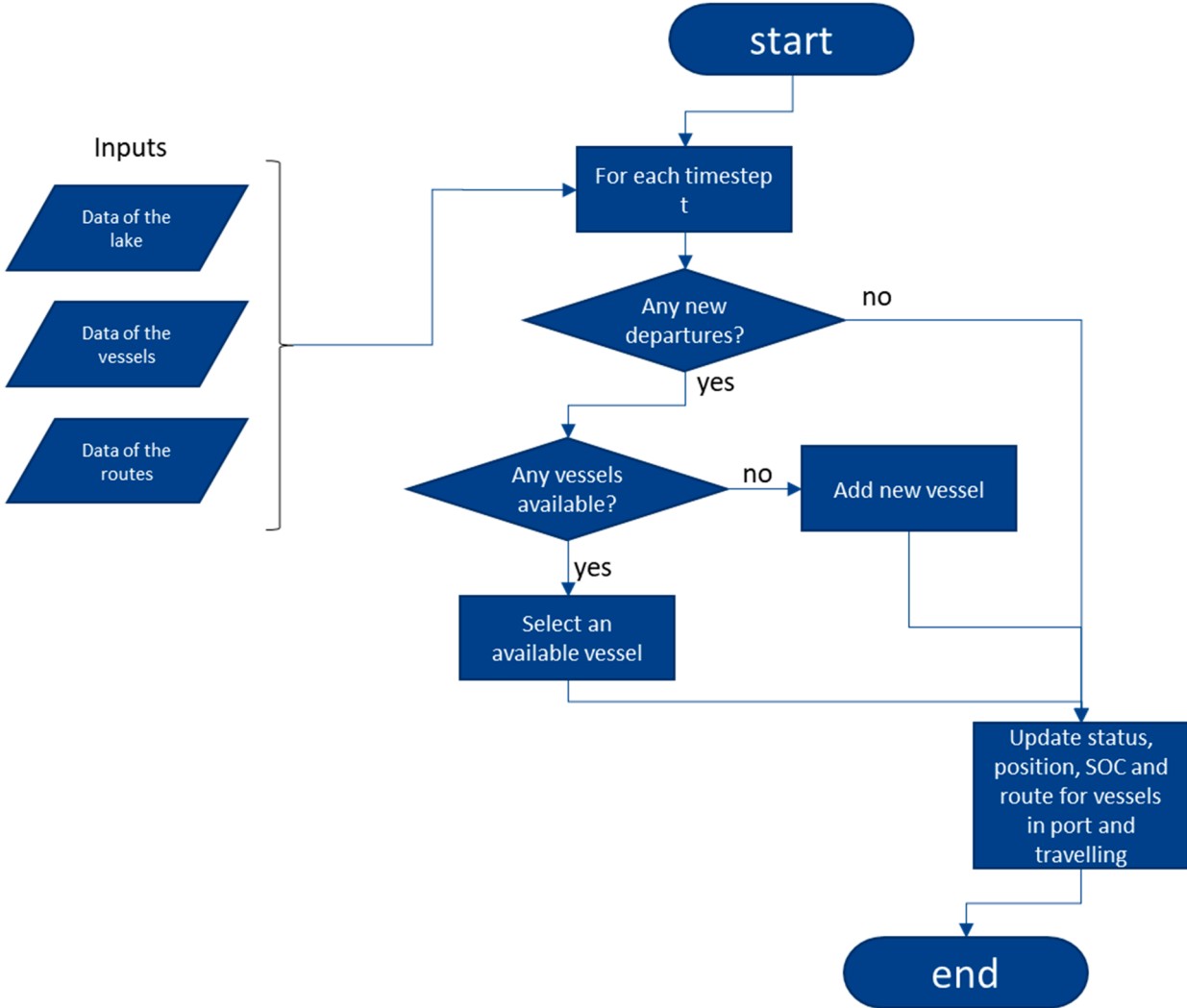

**Figure 1.** Diagram of the model.

The key output of the model is the minimum number of electric vessels required to entirely replace the current diesel fleet by maintaining unchanged the time schedule of vessels arrivals and departures. This reveals how many electric vessels would be needed if making a full transition to a sustainable vessels fleet. In addition, the model outputs the status, position, and state of charge of each individual vessel through time. This provides further critical information:

- Status indicates whether each vessel is "available", "traveling", or "charging" at each timestep. This shows how the vessels are assigned to routes over the schedule.
- Position tracks where each vessel is located on the lake at each moment. This illustrates how the vessels move across the different routes.
- State of charge shows the remaining battery capacity of each vessel through time. This helps identify charging needs and ensures battery levels are sufficient.

The full set of outputs gives a detailed picture of how an optimal electric fleet could be configured and scheduled, delivering key insights into the feasibility and logistics of transitioning away from diesel vessels.

The status and state of charge of each vessel are updated for each timestep and each route. The status of a vessel can be "available", "traveling", or "charging", and the state of charge represents the remaining battery capacity of the vessel.

The model evaluates the minimum number of electric vessels necessary to completely cover the routes that are currently covered by diesel vessels. For each vessel, the model calculates the position, state of charge, and other relevant parameters.

The objective function of the model is presented in Equation (1).

$$Minimize\ Z = \sum_i \sum_j X_{i,\,j} \tag{1}$$

where

- $Z$ is the total number of vessels,
- $X_{i,j}$ is a binary variable that equals 1 if vessel $i$ on route $j$ is used and 0 otherwise.

The model is subject to the following constraints:

1. Each route must be covered by at least one vessel. The constraint is shown in Equation (2).

$$\sum_i X_{i,\,j} \geq 1 \quad for\ all\ routes\ j \tag{2}$$

2. The state of charge of a vessel must be sufficient to cover a route (see Equation (3)), where $S_{i,j}$ is the state of charge of vessel $i$ on route $j$ and $D_j$ is the minimum required state of charge to complete route $j$. $D_j$ is given by the product between the specific energy consumption and the distance of route $j$.

$$S_{i,j} \geq D_j \tag{3}$$

This model provides a practical tool for planning the transition to sustainable inland waterways transport in local lakes. It can be adapted to different settings by adjusting the parameters and constraints according to the specific characteristics of the lake, the boats, and the routes.

The optimization problem in this study can be formulated as a mixed-integer linear program, with the decision variables representing the number of vessels and their assignments to routes, and constraints enforcing requirements like route coverage and vessel charge levels. This problem structure represents the ideal mathematical formulation for finding the globally optimal fleet size and assignments. However, the algorithm used to solve this optimization problem in this article is a greedy heuristic, rather than an exact optimization technique. The greedy heuristic operates by sequentially making local decisions to assign the best available vessel to each route in each time step. While the greedy approach finds feasible solutions quickly, it does not guarantee finding the true global optimum. The tradeoff is between solution quality and computational efficiency. Alternative algorithms like the primal-dual interior-point algorithm [34], the interior point method [35], or the branch-and-bound algorithm [36] could solve different optimization problems exactly but require significantly more computation. The greedy algorithm leverages elements of dynamic programming, keeping track of vessel states over time to enable better sequential decisions.

### 3. Case Study

The case study for this research is Orta lake, a lake in the Piemonte region, which is in the north of Italy. The lake spans an area of approximately 18.2 square kilometers, with a maximum depth of 143 m, making it the ideal location for short route inland waterways transport. The current inland waterways operations on the lake are primarily supported by three diesel-powered vessels with capacity for 100 people, which provide essential transportation for both locals and tourists alike. The lake currently operates a fleet of three

diesel-powered vessels for inland waterway transport, and the objective of this study is to strategize the transition towards an electric-powered fleet.

The first route, as illustrated in Figure 2, commences from Omegna (port 0) in the north of the lake, moving towards the Island of San Giulio before arriving to Orta Piazza (port 1). This route serves as a critical link between different towns and the Island of San Giulio, which hosts a renowned ancient monastery that is a significant attraction for tourists. The second route is a shorter one, running in a circular pattern around the lake, stopping at several small villages and tourist spots along the perimeter. The round-trip journey serves not only as a means of transport but also offers scenic tours around the lake. The vessels operate on a scheduled timetable, with increased frequency during the tourist season. For this reason, this study focuses on the summer touristic season. This decision is rooted in the understanding that the summer months represent a period of significantly heightened inland waterways activity due to an influx of tourists, making it the most critical time of the year in terms of the volume and frequency of connections. By modeling the most demanding scenario in terms of vessel usage, the study seeks to ensure that the proposed sustainable transition not only serves the requirements of the off-peak periods but is also capable of effectively managing the heightened demand during the peak tourist season.

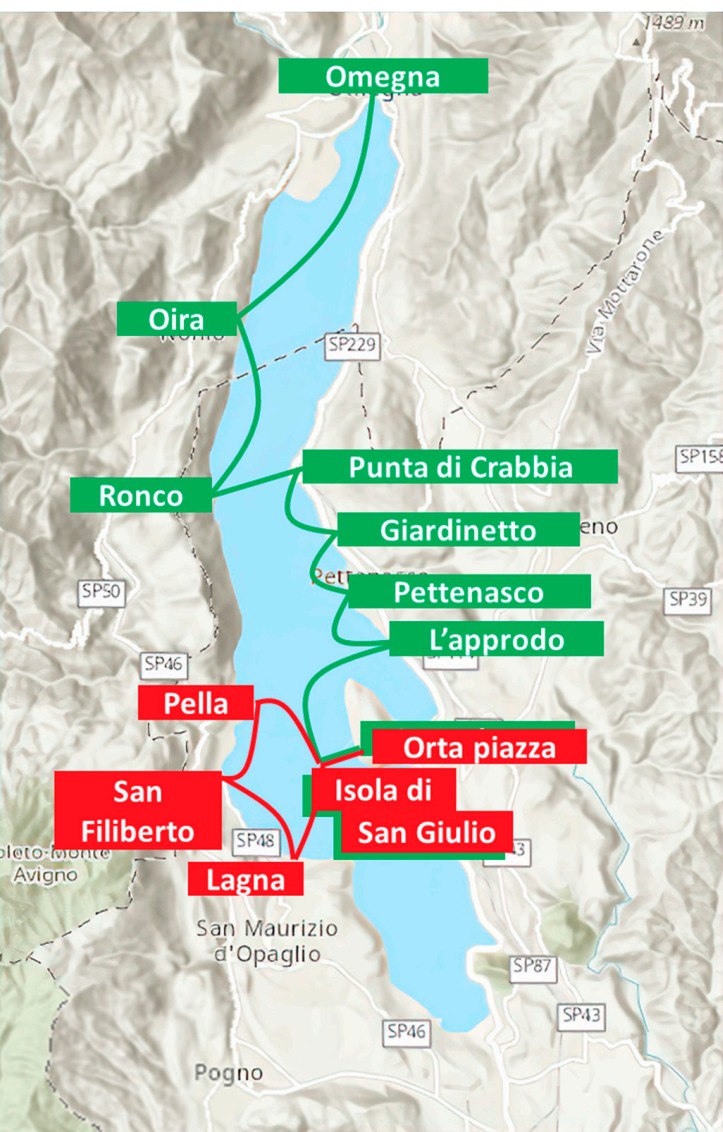

**Figure 2.** Orta lake with the two vessels' routes.

The model operates in discrete timesteps of 5 min. The lake has two main ports, and the distances between these ports are defined as follows (shown in Figure 3):

- The distance between Port 0 (Omegna) and Port 1 (Orta piazza) is 15 km (green line).
- The distance for the round trip at Port 1 (Orta piazza) is 7 km (red line).

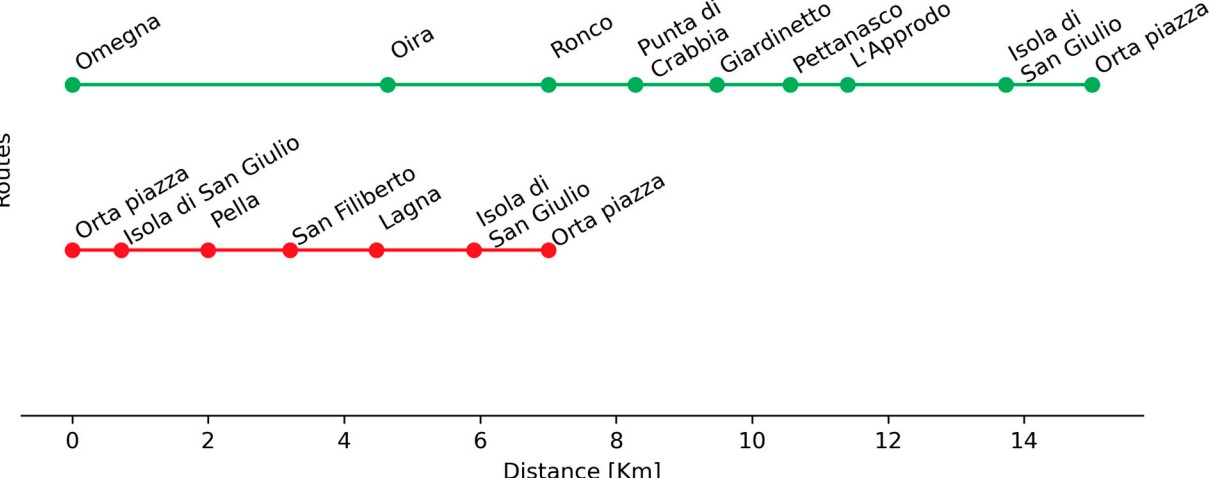

**Figure 3.** Routes considered for the Orta lake case study with intermediate stops and distance.

The connections between the ports, along with their departure times, are defined as follows:

- The connections from Port 0 to Port 1 are at 09:00 and 14:15.
- The connections from Port 1 to Port 0 are at 12:35 and 18:00.
- The round trip connections at Port 1 (red route) are at various times throughout the day, from 10:00 to 18:45.

The electric vessels considered for the transition have the following technical characteristics, taken from the manufacturer website, Fjellstrand [37], and Baird Maritime:

- Battery capacity: 1524 kWh.
- Maximum speed: 42.5 km/h.
- Specific energy consumption: 35.78 kWh/km (this number given by the manufacturer already accounts for the resistance of wind and waves).
- Passenger capacity: 147.

The model uses these inputs to evaluate the minimum number of electric vessels necessary to completely cover the routes and timetables that are currently covered by diesel boats. The results of this evaluation are presented in the next section.

The overall modeling approach of representing the fleet transition challenge as a mixed-integer linear optimization model solved using a greedy heuristic algorithm has precedent from other transportation contexts such as electric buses [26]. Specifically, Sparber et al. [26] developed a model to evaluate scenarios for transitioning a regional bus fleet to zero-emission options, utilizing a greedy algorithm similar to our approach. Their work provides validation that a greedy heuristic can effectively solve optimization problems related to sustainable fleet transition. Additionally, the greedy heuristic provides an approximate but rapid solution, trading off optimality for computational speed. Lastly, our model is validated through the use of real-world data for inputs, including electric vessel characteristics from existing products and current diesel fleet routes and schedules from Orta lake. Anchoring the model in real data produces credible, applicable results.

## 4. Results

This section presents the results obtained from applying the optimization model to the case study of Orta lake. The results are visualized in Figure 4, which shows four subplots of

the state of charge of all necessary boats in four scenarios. These scenarios are characterized by four different charging powers of the charging stations that are assumed to be built in each port: 1000 kW, 500 kW, 200 kW, and 100 kW.

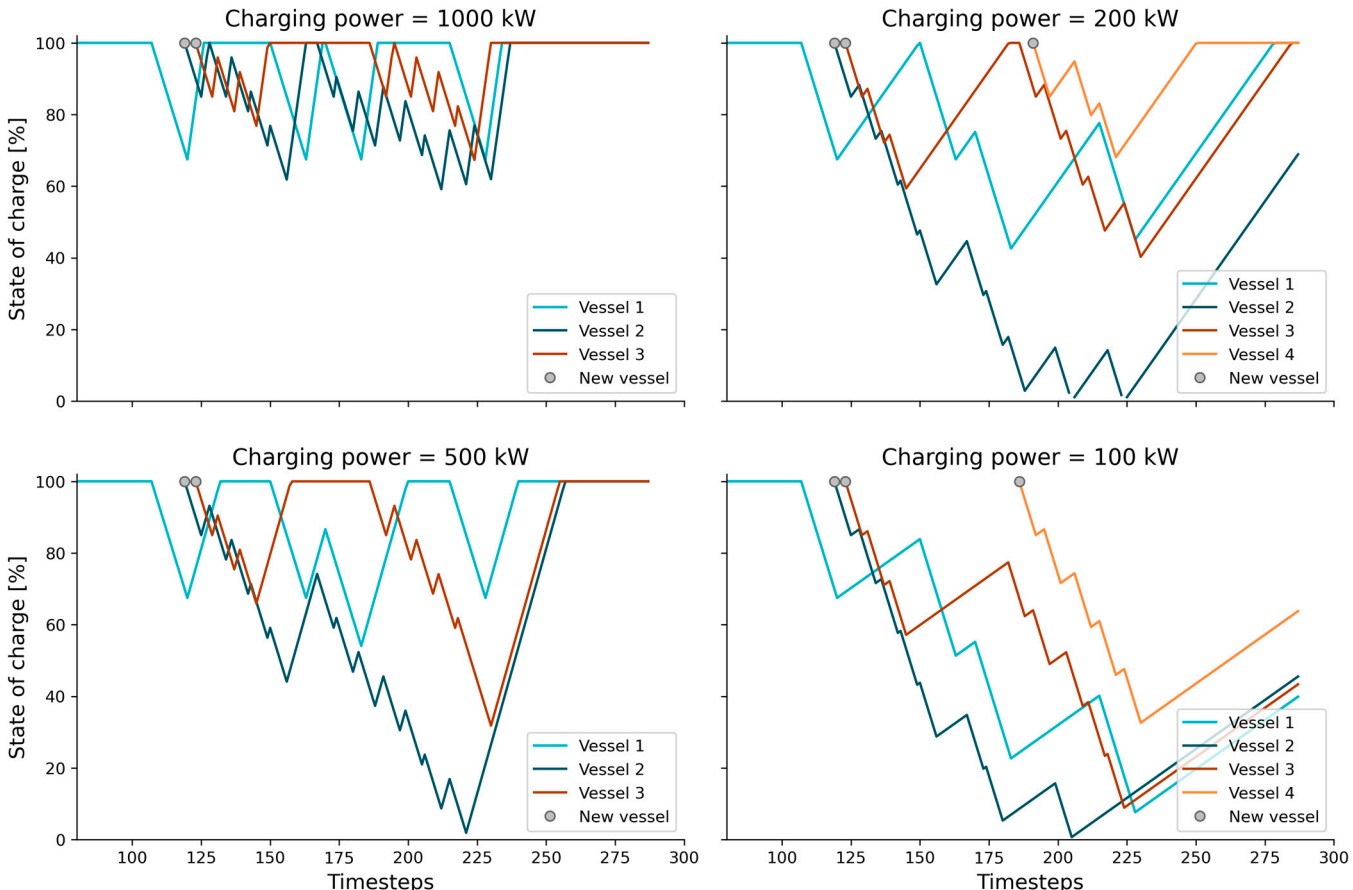

**Figure 4.** Number of boats and their state of charge over 24 h for each charging power scenario (1000 kW, 500 kW, 200 kW, 100 kW).

In the scenario with 1000 kW of charging power, the results show that the entire fleet of boats (3) can be substituted with the same number of electric boats without any change to the routes or to the timetable. Furthermore, the state of charge of the entire fleet never goes lower than 50%, indicating a sufficient buffer for unexpected events or fluctuations in energy consumption.

The scenario with 500 kW of charging power also allows for the substitution of the entire fleet with three electric boats. However, in this case, the state of charge of these boats goes under 50%, reaching very low values. This represents a risk for the safe management of the fleet, as it leaves less room for unexpected events or fluctuations in energy consumption.

In the scenarios with lower charging power (200 kW and 100 kW), the minimum number of vessels increases to four. These scenarios illustrate the impact of the charging rate on the fleet configuration and operation. The lower charging rate of the electric batteries results in less steep charging segments compared to the other charts, indicating slower charging times.

Figure 4 also shows scatter points marked whenever the model adds a new electric vessel to the fleet. These points appear on all four charging power scenario subplots. The reason is that the model initializes with only a single electric boat, then incrementally expands the fleet over the day as needed to meet scheduled departures. Whenever there is a scheduled route departure from a port but no electric vessel with adequate charge is available, a scatter point indicates the model assigning a new electric boat. The scatter points

visually demonstrate the discrete events of new vessel addition across all the scenarios. Since the charging power limitations make the existing vessels insufficient to cover the timetable, the model progressively supplements the fleet at points during the day.

These results provide insights into the feasibility and implications of the transition to sustainable inland waterways transport in Orta lake. They highlight the importance of the charging infrastructure, specifically the charging power of the charging stations, in determining the optimal fleet configuration and operation.

## 5. Discussion

The model developed in this study is based on real data provided by manufacturers of electric vessels available on the market. It operates in discrete timesteps and updates the status and state of charge of each vessel for each timestep and each route. This allows the model to evaluate the minimum number of electric vessels necessary to completely cover the routes that are currently covered by diesel vessels in the summer (the most critical season concerning the frequency of the connections and the number of tourists).

The results obtained from applying the model to the case study of Orta lake highlight the importance of the charging infrastructure in the transition. Specifically, the charging power of the charging stations plays a crucial role in determining the optimal fleet configuration and operation. Higher charging power allows for the substitution of the entire fleet with the same number of electric boats without any change to the routes or to the timetable, while lower charging power requires more electric boats to cover the same routes.

These findings have important implications for the planning and implementation of the transition to sustainable inland waterways transport in local lakes. They suggest that adequate charging infrastructure is a key enabler of the transition and that careful consideration should be given to the charging power of the charging stations when planning the fleet configuration and operation.

The results shown in Figure 4 provide important insights into the fleet configuration and charging needs under different charging power scenarios. In particular, the 500 kW scenario, where only one vessel reaches very low state of charge levels, suggests that more targeted investments in charging infrastructure could be sufficient rather than scaling up all charging points to 1000 kW. For example, installing a 1000 kW charging station at only the port where this one vessel with a low state of charge docks could improve the situation while requiring lower infrastructure costs compared to upgrading all ports to 1000 kW.

## 6. Conclusions

This study presents an optimization model for the transition to sustainable inland waterways transport in local lakes, with a specific focus on Orta lake in Italy. The model aims to determine the minimum number of electric vessels required to replace the existing diesel-based fleet, taking into account various parameters such as capacity, range, charging time, and costs of the available electric vessel models.

The results obtained from applying the model to the case study of Orta lake provide valuable insights into the feasibility and implications of the transition. They highlight the importance of the charging infrastructure, specifically the charging power of the charging stations, in determining the optimal fleet configuration and operation.

In scenarios with higher charging powers (1000 kW and 500 kW), the entire fleet of boats can be substituted with the same number of electric boats without any change to the routes or to the timetable. However, in the scenario with 500 kW charging power, the state of charge of the boats can drop under 50%, representing a risk for the safe management of the fleet. In scenarios with lower charging powers (200 kW and 100 kW), the minimum number of vessels increases to four, indicating the need for more electric boats to cover the same routes. These findings contribute to the growing body of knowledge on sustainable inland waterways transport and offer a practical tool for planning the transition in local lakes or other maritime settings.

## 7. Future Work

The next stage of model development will incorporate optimization of costs. An optimization approach focused on minimizing overall costs could help identify the ideal trade-offs between the number of vessels, the number and placement of charging points, and charging power capacity. The model could be enhanced to determine the optimal locations for charging points and the optimal sizing of charging power at each location based on minimizing the total costs. This cost optimization approach would provide greater insights into the most affordable configurations for transitioning the fleet. It would balance the complex interdependencies between the charging needs, routes, vessel state of charge, and infrastructure requirements. This would support decision-makers in planning affordable, sustainable transitions of inland waterway transport fleets. Further, this model can be scaled up to maritime fleets once suitable technologies are identified and commercially available.

**Author Contributions:** Conceptualization, M.G.P., A.Z. and W.S.; methodology, M.G.P. and W.S.; software, M.G.P.; formal analysis, M.G.P.; resources, M.G.P., A.Z., A.G., G.R. and W.S.; data curation, M.G.P. and G.R.; writing—original draft preparation, M.G.P.; writing—review and editing, M.G.P., A.Z., A.G., G.R. and W.S.; visualization, M.G.P.; supervision, A.Z. and W.S.; project administration, W.S.; funding acquisition, A.Z. and W.S. All authors have read and agreed to the published version of the manuscript.

**Funding:** This research was funded by MOBSTER—IV Avviso project, ID 3846117. MOBSTER is a project co-funded by the European Union, the European Regional Development Fund, the Italian Government, and the Swiss Confederation and Cantons as part of the Interreg V-A Italy-Switzerland Cooperation Programme.

**Institutional Review Board Statement:** Not applicable.

**Informed Consent Statement:** Informed consent was obtained from all subjects involved in the study.

**Data Availability Statement:** Publicly available datasets were analyzed in this study. These data can be found here: https://www.navigazionelagodorta.it/it/index.php, accessed on 1 May 2023, https://www.bairdmaritime.com/work-boat-world/passenger-vessel-world/ferries/vessel-review-medstraum-electric-catamaran-ferry-for-norways-kolumbus/, accessed on 1 May 2023.

**Acknowledgments:** The authors would like to thank VCO Trasporti S.r.l. for providing boat routes and timetables and the Interreg V-A Italy-Switzerland Cooperation Program for funding the project MOBSTER.

**Conflicts of Interest:** The authors declare no conflict of interest.

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
