# Peer review of "Optimal Fleet Transition Modeling for Sustainable Inland Waterways Transport"

_applsci, doi:10.3390/app13179524_

Round 1

Reviewer 1 Report

1) Provide a list of contributions. It is not clear where and how the submitted manuscript introduces novelty. 

2) How can one ensure that the optimization problem (1)-(3) is convex; that is, it has one and unique optimal solution. 

3) The authors need to comment on how one can solve the optimization problem (1)-(3). Possible ways are 1) primal–dual interior-point method [R1], 2) Robust to Early Termination Method [R2], and 3) Interior Point method [R3], which should be mentioned in the manuscript. 

[R1]. "A primal-dual interior-point algorithm for nonsymmetric exponential-cone optimization", 2022. [https://doi.org/10.1007/s10107-021-01631-4]

[R2]. "ROTEC: Robust to early termination command governor for systems with limited computing capacity", 2022. [https://doi.org/10.1016/j.sysconle.2022.105142]

[R3]. "Interior point method for dynamic constrained optimization in continuous time", 2016. [http://doi.org/10.1109/ACC.2016.7526550]

4) Consider comparing your method with a state-of-the-art method. This would help the readers to understand pros and cons of the method. 

Acceptable. 

Author Response

Please find attached the detailed answer to reviewer's comments

Reviewer 2 Report

(1) How the proposed research work is validated over existing techniques?

(2) What are the basic steps which are used to establish this research work?

(3) What was the motivations for proposing the techniques.

(4) More explanation on presented results is expected.

(5) The sequence of the paper should be- (i) introduction, (ii) Related works, (iii) materials and methods, (iv) results, (v) discussion, (vi) conclusion, and (vii) future work

(6) Author should add contributions in the last of the introduction or in the last of materials and methods section.

(7) All tables and figures should be explained clearly.

(8) Author should add future scope of the paper.

(9) Add important flow charts of the work.

(10) What is the computational complexity of the proposed technique? Author must compare with the existing techniques.

(11) Author must check the english of the paper in the presence of native English speaker.

After making these mandatory changes, the manuscript can be published.

Author Response

(The authors gave the same response as above.)
